# Atomic level fluxional behavior and activity of CeO$_2$-supported Pt catalysts for CO oxidation

Joshua L. Vincent [1] & Peter A. Crozier[1 ✉]

Reducible oxides are widely used catalyst supports that can increase oxidation reaction rates by transferring lattice oxygen at the metal-support interface. There are many outstanding questions regarding the atomic-scale dynamic meta-stability (i.e., fluxional behavior) of the interface during catalysis. Here, we employ aberration-corrected *operando* electron microscopy to visualize the structural dynamics occurring at and near Pt/CeO$_2$ interfaces during CO oxidation. We show that the catalytic turnover frequency correlates with fluxional behavior that (a) destabilizes the supported Pt particle, (b) marks an enhanced rate of oxygen vacancy creation and annihilation, and (c) leads to increased strain and reduction in the CeO$_2$ support surface. Overall, the results implicate the interfacial Pt-O-Ce bonds anchoring the Pt to the support as being involved also in the catalytically-driven oxygen transfer process, and they suggest that oxygen reduction takes place on the highly reduced CeO$_2$ surface before migrating to the interfacial perimeter for reaction with CO.

---

[1] School for Engineering of Matter, Transport, and Energy, Arizona State University, Tempe, AZ 85281, USA. ✉email: crozier@asu.edu

Reducible oxides such as $CeO_2$ are widely used catalyst supports due to their ability to undergo rapid and reversible oxygen uptake and release[1–3]. Additionally, reducible oxide supports offer strong metal-support interactions and can enhance the rates of oxidation reactions by transferring their lattice oxygen to reactive adsorbates at the metal-support interface[4–9]. For CO oxidation, the interfacial oxidation process is typically described in terms of a Mars-van Krevelen mechanism[10,11], in which $CeO_2$ lattice oxygen is transferred to CO at the perimeter of the metal-support interface, which is also called the three-phase boundary. There is very little information about the atomic structure and structural dynamics of an active metal-support interface performing catalysis. Consequently, there are many outstanding questions regarding the atomic-scale evolution of the interface, metal particle, and adjacent oxide surface during catalysis. For example, how does the metal particle and the metal-support interface behave during catalysis if oxygen vacancies are constantly being created and annihilated? Does such a process destabilize the metal particles and make coarsening more likely? Where is the likely site for molecular oxygen reduction and reincorporation into the lattice, which completes the catalytic cycle?

One may expect substantial changes in metal particle shape and bonding to occur during catalytic turnover, as the interfacial energy fluctuates from the continual creation and annihilation of oxygen vacancies. The adhesion between the metal particle and support may also weaken significantly since bridging interfacial oxygens are responsible for anchoring the metal to the support[12–16]. Oxygen vacancies will introduce strain in the $CeO_2$, which could further destabilize the system and make it more reactive[17]. Recently, there has been an emerging paradigm that has roots in surface science[18–20] and chemistry[21] for understanding catalytically active sites as dynamic, meta-stable, or so-called fluxional species[22–26]. In order to deepen our understanding of the factors affecting catalysis and to develop strategies for improved catalyst design, it is essential to elucidate and describe the structural meta-stability (i.e., fluxional behavior) that occurs at the atomic-level during simultaneous catalytic turnover.

Aberration-corrected in situ environmental transmission electron microscopy (AC-ETEM) is a powerful tool capable of providing atomic-level information about dynamic structures and how they evolve during catalysis. In the last decade, there have been considerable advances for studies under reaction conditions, including the development of in-line gas analysis (e.g., electron energy-loss spectroscopy and/or mass spectrometry). Measurements of the gas composition can confirm that catalysis is actually taking place, and in favorable cases they may be used to quantify the in situ conversion. Recent investigations have used AC-ETEM along with in situ microreactors coupled to in-line spectroscopy or downstream mass spectrometry in order to correlate observations of catalyst structure with changes in gas composition or conversion[27–31]. While impressive, these and other investigations have often involved the observation of larger 10–100 nm metal nanoparticles that are dispersed on non-reducible supports (e.g., $SiO_2$) or on $SiN_x$ films. In order to understand Mars-van Krevelen oxidation, however, the metal nanoparticles must be supported on a reducible oxide, since strong metal-support interactions play a significant role in the reaction mechanism. To our knowledge, no such observations have been reported.

Additionally, in order to establish a truly *operando* connection between the observed structure and the catalyst's activity, it is necessary to move beyond simple in situ measurements of reactant conversion and directly correlates the observations with the reaction rate of the working catalyst, as first discussed by Bañares et al.[32,33]. It is important to stress that an in situ measurement of conversion is not itself and in most cases is not directly related to the reaction rate

of the working catalyst. In the field of reaction engineering, this knowledge has led to the design of controlled catalytic reactors operated under easily modeled conditions wherein the conversion can be directly related to the kinetics of the catalyzed chemical reaction. The in situ reactors used for ETEM catalysis research are complex but may nonetheless be modeled using, e.g., finite element approaches, enabling the determination of quantitative catalytic reaction rates during atomic-level structural observation[34].

In the current work, we use *operando* TEM to explicitly and quantitatively correlate the catalytic reaction rate with two forms of structural dynamics processes. The Mars-van Krevelen mechanism is intimately associated with metal-support interactions, and we provide direct observations and characterization of the interfacial behavior over a range of different activities. Finite element simulations of the *operando* ETEM reactor are employed to develop and support the chemical reaction rate analysis. Structural dynamics occurring at/near the sites that comprise the metal-support interface are correlated with the catalyst's turnover frequency for CO oxidation. Additionally, the catalyst is observed at 300 °C in a non-reactive atmosphere of $N_2$ in order to rule out temperature-induced fluxionality from that associated with catalysis. Uniquely, during CO oxidation, the increasing frequency of catalytic turnover is seen to correlate with an increasing concentration of $CeO_2$ surface oxygen vacancies and dynamic structural behavior marking an enhanced rate of oxygen vacancy creation and annihilation; at the same time, the ~1.5 nm Pt nanoparticles become increasingly destabilized and undergo continuous and more frequent fluxional behavior. The *operando* electron microscopy approach described here should be applicable to a large number of nanoparticle catalysts, which will enable the identification of catalytically functional surface structures and strengthen our ability to establish (dynamic) structure-activity relationships.

## Results

**Catalyst morphology and ex situ characterization.** The catalyst's activity for CO oxidation was evaluated in a packed-bed plug-flow reactor. The light-off curves for the bare and Pt-loaded $CeO_2$ are shown in Fig. 1a. An identical mass loading of catalyst was used for both cases. The activity of the bare $CeO_2$ support (blue) is demonstrably less than the Pt-loaded $CeO_2$ (red) and the effect of the blank reactor (black) is negligible at working temperatures. No difference in the $CeO_2$-supported Pt catalyst's activity was detected on the temperature ramp-up to full conversion (filled red triangles pointing up) in comparison to the ramp back down to room temperature (empty red triangles pointing down). An Arrhenius analysis of the light-off curves for the Pt/$CeO_2$ catalyst (Supplementary Fig. 3) shows linear behavior, with an apparent activation energy of $E_{a,\ app} = 74\,\mathrm{kJ\,mol^{-1}}$ calculated from the linear regression, which is consistent with activation energy values reported previously[6].

The powder XRD patterns of the bare $CeO_2$ and Pt-loaded $CeO_2$ are shown in Supplementary Fig. 1. A simulated XRD pattern of an infinite crystal of $CeO_2$ (*JCPDS* No. 34-0394) matches well with the experimental XRD patterns of the catalyst, indicating that the sample is phase-pure $CeO_2$ (space group Fm-3m, $a = 5.41$ Å) along with highly dispersed, small Pt nanoparticles.

A HAADF-STEM image of a typical Pt-loaded $CeO_2$ nanoparticle is shown in Fig. 1b. The Pt nanoparticles are well-dispersed on the $CeO_2$ nanoparticle support and are around 2 nm in size. The size distribution of the Pt nanoparticles was further investigated and quantified with HAADF-STEM imaging. Details are provided in Supplementary Fig. 2 of the Supplementary Information. Many different regions of the catalyst sample were

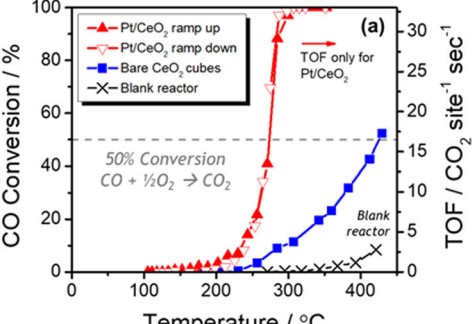
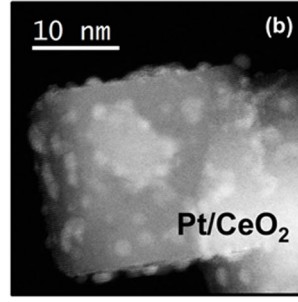
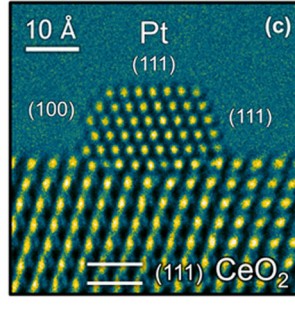

**Fig. 1 Activity and structural characterization of Pt/CeO₂ catalyst. a** Catalytic activity for CO oxidation was evaluated in a packed-bed plug-flow reactor. The activity of the bare CeO₂ support is demonstrably less and the effect of the blank reactor is negligible. **b** Z-contrast STEM image of a typical Pt-loaded CeO₂ nanoparticle showing the dispersion of roughly 1.6 nm Pt nanoparticles loaded on the CeO₂ nanoparticle support. **c** HRTEM image of a typical Pt nanoparticle supported on a (111) surface of a CeO₂ nanoparticle, with indices of exposed Pt surfaces marked in the image.

imaged and the diameter sizes of 475 Pt nanoparticles were measured, revealing that the average Pt nanoparticle size was 1.6 nm. Quantifying the Pt nanoparticle size allowed for catalytic turnover frequencies (TOFs) to be determined on an interfacial perimeter-site basis. A derivation and sample calculation are provided in the Supplementary Information.

Figure 1c displays an HRTEM image of a typical CeO₂-supported Pt nanoparticle. The Pt nanoparticles are stable and well-facetted when imaged ex situ, with the one shown in Fig. 1c exposing nanosized (100) and (111) surfaces, which is typically observed. The Pt nanoparticles exhibit a well-defined epitaxy with the support, with virtually all nanoparticles presenting (111) surfaces to the metal-support interface, which in this case is comprised of a CeO₂ (111) surface on the support side. Well-defined orientation relationships have been reported for CeO₂-supported Pt nanoparticles, and they can be attributed to epitaxial relationships that indicate strong structural interactions between the metal and the support[2,35,36]. The Pt particles remained stable during the HRTEM observation at room temperature.

**In situ ETEM imaging under CO oxidation reaction conditions.** The catalyst was imaged in situ under CO oxidation reaction conditions. Figure 2 shows an in situ ETEM image time-series of a typical CeO₂-supported Pt nanoparticle in 0.57 Torr of CO and O₂ at 144 °C. The Pt particle occupies a short CeO₂ (111) nanofacet and each side of the particle is in contact with a CeO₂ (111) step edge. Images of the catalyst obtained in situ under reaction conditions show prominent motion artifacts and features attributable to fluxional behavior. The time-series is comprised of four images acquired in succession, each with a 0.5 s exposure time. Figure 2a displays the time-averaged image obtained by summing together the individual frames over the entire [0–2.0] s acquisition period. Relative to the bulk of the underlying CeO₂ support, which appears with stable and well-defined white atomic column contrast, the Pt nanoparticle presents white as well as black atomic column contrast. The weak and varying Pt contrast in the 2.0 s time-averaged image of Fig. 2a is not a consequence of poor microscope imaging conditions, as evidenced by comparison with the bulk CeO₂. Rather, the Pt particle is demonstrating fluxional behavior, in which the nanoparticle dynamically progresses through a series of reconfigurations over the course of the time-averaging period. Multislice TEM image simulations have been performed to investigate this effect (see Supplementary Fig. 7). The simulations demonstrate the appearance of mixed black and white Pt atomic column contrast when the particle is aligned with the incident beam and imaged at an electron-optical defocus of 2 nm. Furthermore, the simulations show that contrast

reversals can occur when the Pt nanoparticle tilts by a few degrees, e.g., due to a rigid-body rotation.

For the sake of clarity in the following discussion, we define three locations of reference, namely (1) the free CeO₂ surface, which is comprised of the sites at the freely exposed CeO₂ surface around the Pt nanoparticle, (2) the three-phase boundary, which is comprised of the sites at the perimeter of the metal-support interface, and lastly (3) the buried interface, which is comprised of the sites within the metal-support interface that are not exposed to the gas phase. In the 2.0 s time-averaged image (Fig. 2a), diffuse contrast is observed at the three-phase boundary (right arrow) and along the free CeO₂ surface on the CeO₂ (111) terrace to the left of the Pt nanoparticle (left arrow). In comparison, the subsurface and bulk Ce sites show sharp and clearly resolved atomic columns. The diffuse contrast or local streaking/blurring observed at the CeO₂ surface and Pt/CeO₂ interface is therefore not due to drift in the hot stage or the electron optics, but instead arises from dynamic structural reconfigurations occurring at these specific sites[25].

An examination of the in situ image time-series reveals the atomic-scale structural dynamics and how they evolve over time. Figures 2b–2e show the four individual 0.5 s exposure frames taken over the [0–2.0] s acquisition period (the images have been processed with a bandpass filter for clarity). Figure 2(f1)–2(f4) displays the Fourier transform (FT) at each time interval, taken from the windowed region around the Pt nanoparticle that is indicated by the dashed box in Fig. 2b. An analysis of the major spots that appear in the FTs indicates the presence of metallic Pt (see Supplementary Fig. 8). (The FTs were produced from unfiltered images that were pre-processed with a 2D Hanning function to suppress edge artifacts caused by windowing).

As seen both in the time-series images and corresponding FTs, the Pt nanoparticle undergoes a series of structural reconfigurations that involve dynamic restructuring at the Pt particle surface and at the Pt/CeO₂ interface. At the same time, certain surface and interfacial Ce sites exhibit dynamic events that occur along with those observed in the supported Pt. In Fig. 2c, we observe the catalyst at a moment of relative structural stability, evidenced by the surface faceting and atomic column contrast in the Pt. In Fig. 2c, the arrowed Ce atomic columns at/near the three-phase boundary appear with diffuse contrast but are clearly resolved. In the next frame, 0.5 s later (Fig. 2d), the same arrowed Ce sites have either disappeared or substantially blurred. The Pt has undergone a rotation or tilt, evidenced by the diagonal streaking in the Pt and by the loss of associated Bragg spots in the FT shown in Fig. 2(f3). Another 0.5 s later (Fig. 2e), the Pt particle shows weak contrast with no evident lattice fringes. The particle has also adopted an apparently rounded shape, which is marked with a dashed line in

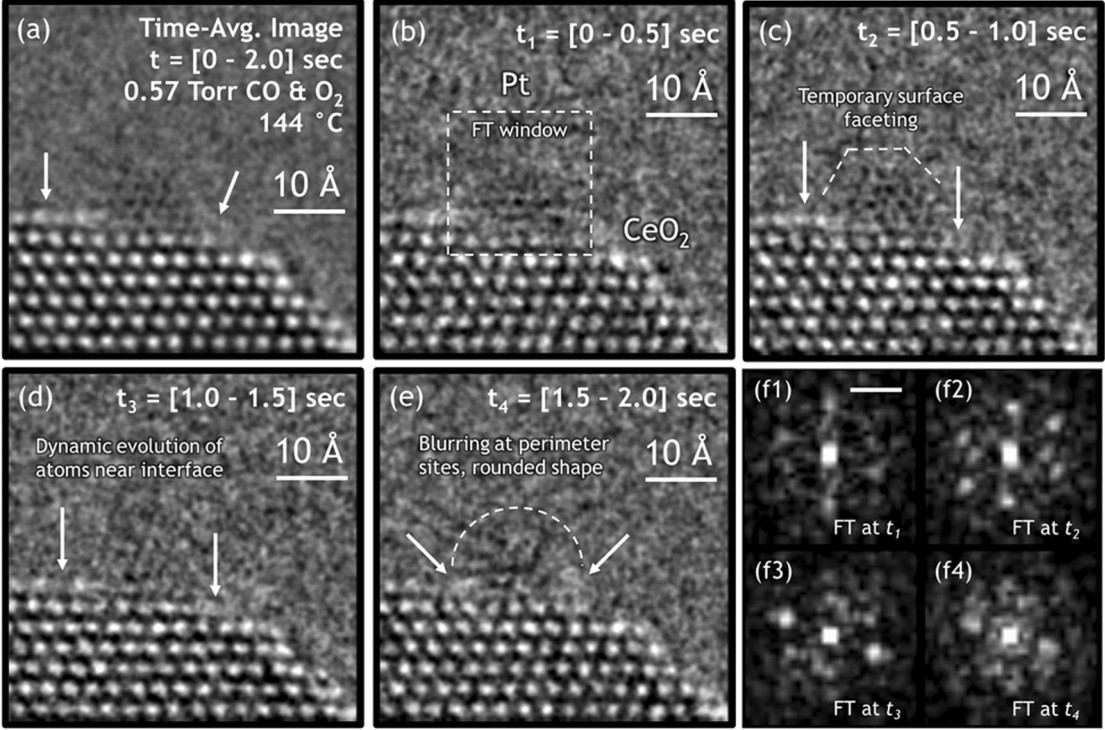

**Fig. 2 Fluxional behavior observed under CO oxidation reaction conditions.** In situ ETEM image time-series of $CeO_2$-supported Pt NP at 144 °C in 0.57 Torr of CO and $O_2$. **a** Time-averaged image of the catalyst, obtained by summing together the individual 0.5 s exposure frames over the entire [0–2.0] s acquisition period. **b–e** Atomic-scale structural dynamics that evolve over 0.5 second intervals from $t = 0$ s to $t = 2.0$ s. **f1–f4** FT taken at each time interval from the windowed region around the Pt NP, as denoted in (**b**). The scale bar in (**f1**) is 5.0 nm-1. Images have been processed with a bandpass filter for clarity. FTs were produced from unfiltered, windowed images that were processed with a Hanning function to remove edge artifacts caused by windowing; the modulus of the FT is shown.

the image. The lack of lattice fringe contrast in the Pt signifies that the supported particle is undergoing dynamic restructuring at a speed that is much faster than the frame exposure time. Interestingly, in this time interval, the Ce sites at the free $CeO_2$ surface and at the three-phase boundary also appear very dynamic. As seen in Fig. 2e, a build-up of bright but diffuse intensity appears at the perimeter of the Pt/$CeO_2$ interface, which is indicated with the arrows. Neighboring Ce sites now show vacancies, so the bright and diffuse contrast at the interfacial perimeter may indicate that nearby surface atoms have migrated to the perimeter sites, where they continue to undergo rapid dynamics that cannot be resolved with the present frame rate (see, e.g.[37]).

At this condition, the reactor temperature is below the light-off point of the catalyst, and no conversion is detected yet in the ETEM cell. While the ensemble of catalyst particles loaded in the ETEM reactor is not yet active enough to produce a measurable conversion, the Pt/$CeO_2$ catalyst still undergoes a variety of dynamic structural reconfigurations which constantly evolve over time. Extrapolating the reaction rates experimentally measured at higher temperatures (and presented in Fig. 3) to this temperature, 144 °C, results in a turnover frequency on the order of $5 \times 10^{-5}$ molecules CO site$^{-1}$ sec$^{-1}$, showing that continuous catalytic turnover is not occurring at an appreciable rate. As such, the dynamic structural reconfigurations observed in Fig. 2 are associated with intermediate steps in the catalytic reaction that have a lower activation energy than the rate-limiting step. Recent density functional theory calculations indicate that abstraction of lattice oxygen by CO on $CeO_2$ (111) surfaces proceeds with an activation energy barrier around 0.4 eV[38].

On the other hand, it is generally well known that the molecular dissociation and gas-phase exchange of $O_2$ is energetically much more challenging, with theoretical calculations

and experimental isotopic oxygen exchange measurements placing the activation energy over $CeO_2$ around 1.1 eV[39,40]. Hence, here we speculate that the observed fluxional behavior is associated with the energetically more facile steps of lattice oxygen abstraction by CO, and that the absence of measurable conversion is due to the lack of oxygen vacancy back-filling from molecular $O_2$ exchange. This fluxional behavior was not observed at room temperature in a vacuum. Additional imaging experiments carried out in an $N_2$ atmosphere at a temperature of 300 °C show a lack of similar structural dynamics, further demonstrating that the fluxional behavior is not due simply to thermal effects (see SI Section 7).

Finally, it is worth noting that the dynamic behavior exhibited by the $CeO_2$-supported Pt nanoparticle in Fig. 2 is typical of the many observed during the ETEM experiment. Additional images and analysis from different particles, for example, are shown in Supplementary Fig. 12. We have also carried out in situ TEM imaging of $CeO_2$-supported Pt nanoparticle catalysts in atmospheres of CO[37], CO and $O_2$ [41], and CO and $H_2O$[26], and shown that substantial fluxional behavior can occur in these reactant gases – even at room temperature. In order to identify structural dynamics unambiguously associated with catalytic chemistry, we proceed to image the catalyst under *operando* conditions and correlate the observed fluxional behavior directly with the chemical kinetics measured in the ETEM cell.

**Quantifying the catalytic reaction rate in the ETEM reactor.** The catalyst was heated until detectable conversions of CO into $CO_2$ occurred. The catalytic reaction rate in the ETEM reactor was determined by experimental EELS measurements supported by finite element simulations. Figure 3a displays a set of

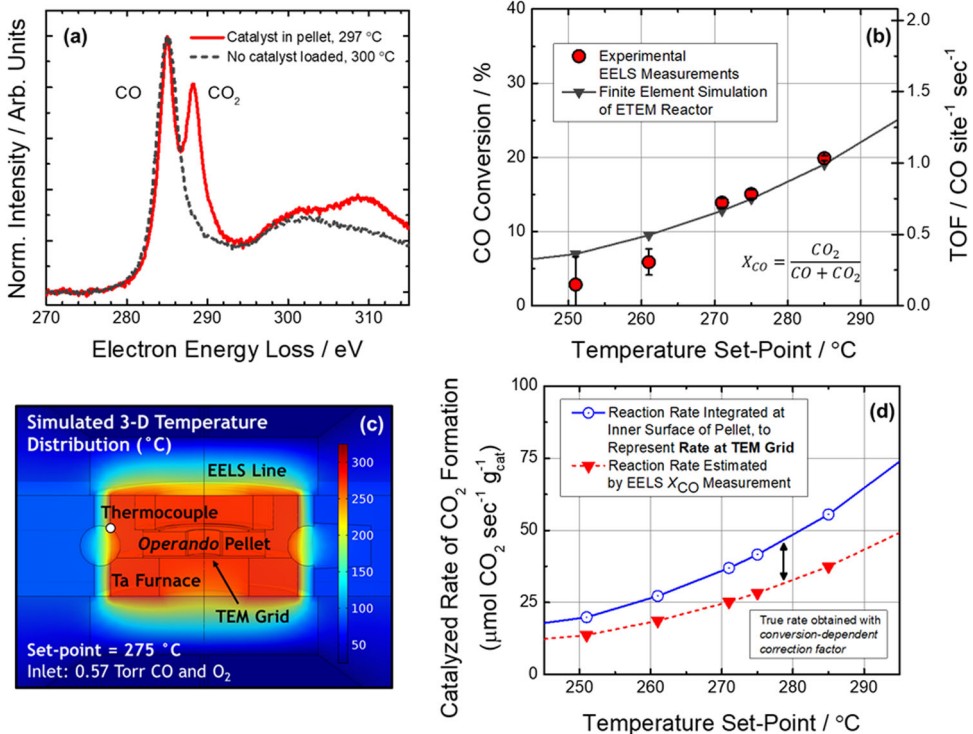

**Fig. 3 Detection and quantification of in situ CO conversion and catalytic reaction rate in the ETEM reactor. a** A set of background-subtracted electron energy-loss spectra taken from the gas atmosphere around the reactor demonstrates that the conversion of CO (C π* peak at 285 eV) into $CO_2$ (C π* peak at 288.3 eV) is attributed unambiguously to the $Pt/CeO_2$ catalyst, not the reactor. **b** The CO conversion detected with EELS during the *operando* ETEM experiment was quantified and plotted as a function of temperature (red circles). The solid gray curve displays the CO conversion evaluated under nominally identical conditions within a finite element simulation of the ETEM reactor. Error bars are given as the standard deviation in five consecutive measurements. **c** The simulated temperature distribution in the model reactor is plotted for a furnace set point of 275 °C. **d** Analysis of the catalyzed rate of product formation in the model, which provides a conversion-dependent correction factor for quantitatively determining the catalytic reaction rate from the in situ conversion measurement.

background-subtracted and normalized energy-loss spectra to demonstrate the detection of catalytically-produced $CO_2$. When no catalyst is loaded in the reactor, the spectrum of gas around the reactor (dashed gray curve) shows a prominent C π* peak at 285 eV, which corresponds to CO. The conversion of CO to $CO_2$ is observed only after $Pt/CeO_2$ catalyst is present (solid red curve). In this case, a second prominent peak appears at 288.3 eV which corresponds to the C π* peak of $CO_2$. Both spectra were acquired under nominally identical conditions for a furnace set point of 300 °C in 0.57 Torr of CO and $O_2$.

The composition of CO and $CO_2$ present in the spectrum can be quantified to calculate the CO conversion[42,43,44]. Figure 3b presents the EELS CO conversions detected during the *operando* ETEM experiment, plotted as a function of the reactor temperature (red circles). At 251 °C, the CO conversion was calculated to be 2.8 ± 3.7%, increasing at 275 °C to a value of 15.1 ± 0.5% and further at 285 °C to a value of 19.8 ± 0.3%. Finite element simulations of the ETEM reactor were performed to establish a framework for linking the CO conversion measured along the EELS line to the reaction rate of the catalyst imaged on the TEM grid. Further details on the simulations are given in the Supplementary Information and in[34]. Steady-state calculations were done under conditions nominally identical to the current experiment, and the gas and temperature distributions were determined during catalysis. The simulated CO conversion measured along the EELS line in the model is plotted as the solid gray curve in Fig. 3b. The simulated conversion curve shows reasonable agreement with the values measured experimentally.

It is important to investigate the extent of any thermal gradients that may exist within the reactor. Figure 3c shows the simulated 3-dimensional temperature distribution in the *operando* ETEM reactor for a furnace thermocouple set point of 275 °C. The temperature distribution appears largely uniform throughout the reactor, and the temperature of the reactor matches well with the furnace set point. The temperature distribution shown in Fig. 3c is representative of the behavior observed at other working temperatures (see, e.g., Supplementary Fig. 5). The high degree of thermal uniformity suggests that the reactor can be treated as approximately isothermal. A quantitative comparison of the temperature difference between the furnace set point and the TEM grid (see Supplementary Table 2), reveals that the typical discrepancy between these two locations is at most 1.6 °C, supporting this assumption.

Next we investigate how the EELS CO conversion measurement can be used to derive quantitative information about the reaction rate of catalyst particles on the TEM grid. Experimentally, the catalytic reaction rate (i.e., the rate of product formation, $r_{CO_2}$, with units of mol $CO_2$ per second) may be estimated from the EELS CO conversion measurement, $X_{CO}$, by multiplying the CO conversion with the known inlet molar flow rate of CO, $\dot{n}_{CO,in}$:

$$r_{CO_2} = \frac{X_{CO} \times \dot{n}_{CO,in}}{m_{cat}} \qquad (1)$$

A common convention in the catalysis literature is to normalize the catalytic reaction rate by the mass of catalyst in

the reactor, $m_{cat}$. The mass-normalized $r_{CO_2}$ estimated from the EELS CO conversion (i.e., Eq. 1) may also be simulated in the model, since it is possible to replicate the conversion measurement by integrating the gas composition along the electron beam path (see SI or[34] for an extended discussion). The simulated $r_{CO_2}$ estimated from the EELS CO conversion is plotted as a function of temperature as the dashed red curve in Fig. 3d.

The rate estimated from the conversion (i.e., Eq. 1) should not be taken as the rate of catalyst particles on the TEM grid. Previously we have shown that the difference between the estimated rate and the true rate at the grid can vary significantly as a function of conversion and grow beyond 200% in some cases, clearly demonstrating the need for and power of a model in determining quantitative chemical kinetics. In the model, the true rate of product formation may be found by integrating the reaction rate throughout the domain of the pellet. The rate may be normalized to mass by also integrating the mass distribution within the same domain. As explained in the SI and in Vincent et al. (2020), the mass-normalized rate for the particles on the TEM grid can be simulated by integrating the mass and rate along the innermost surface of the pellet, where the composition and temperature are both nearly identical to that of the TEM grid[34]. This quantity is plotted as a function of temperature as the solid blue curve in Fig. 3d. Observe that the estimated and true rates are not equivalent, which underscores the importance of establishing a model in order to relate the in situ conversion measurement to the true activity of the imaged catalyst for quantitative *operando* TEM. An analysis of the ratio of the two rates (see Supplementary Table 1) provides a conversion-dependent correction factor that allows the true rate to be calculated correctly from the rate estimated through the EELS conversion measurement.

Finally, we choose to normalize the reaction rate on an interfacial perimeter site-basis, yielding an ensemble in situ TOF. As discussed, a relationship has been derived to link catalyst mass with the average number of Pt perimeter sites (see SI). Thus, the corrected in situ reaction rate can be used to estimate an ensemble in situ TOF for catalyst particles on the TEM grid. This quantity is plotted along the right axis of Fig. 3b. Note that the plotted TOF values were obtained after correcting the rate measured with EELS according to the finite element analysis presented above. At 251 °C, the average in situ TOF was calculated to be 0.15 CO site$^{-1}$ sec$^{-1}$, increasing at 275 °C to a value of 0.80 CO site$^{-1}$ sec$^{-1}$ and further at 285 °C to a value of 1.05 CO site$^{-1}$ sec$^{-1}$.

**Fluxional dynamics occurring during catalysis**. With a quantitative measure of the chemical reaction rate, we proceed to image the catalyst at varying degrees of activity and then correlate the measured reaction kinetics directly with atomic-resolution observations of the dynamic working catalyst structure. Figure 4 displays a set of three 12.5 s time-averaged *operando* ETEM images of the same CeO$_2$-supported Pt nanoparticles taken at (a) 144 °C, where the TOF was determined to be 0.00 CO site$^{-1}$ sec$^{-1}$, (b) at 275 °C, where the TOF was measured to be 0.80 CO site$^{-1}$ sec$^{-1}$, and (c) at 285 °C, where the TOF was measured to be 1.05 CO site$^{-1}$ sec$^{-1}$. The roughly 1.5 nm Pt nanoparticles are supported on small ~2 nm CeO$_2$ (111) nanofacets, with either side of each nanoparticle in contact with CeO$_2$ (111) step edges. We note that the signal-to-noise (SNR) on individual 0.5 s frames is rather poor. The frame averaging approach improves the SNR but also shows the presence of fluxional behavior due to blurring and loss of contrast. For the analysis approach we employed here, the higher SNR present in the averaged image enables a more accurate quantification of the fluxionality. (However, the fluxionality is happening on a timescale that is comparable with the turnover frequencies, as shown Fig. 2 and

Supplementary Fig. 12 with 0.5 s time resolution). We have imaged this catalyst over several orders of electron beam dose rate (and magnification) under a variety of gas conditions. The fluxional behavior observed in the Pt nanoparticles is present even at electron dose rates as low as 60 e$^-$ Å$^{-2}$ s$^{-1}$, so we do not think the electron beam is significantly affecting the observations in this case.

The increasing frequency of catalytic turnover is seen to coincide with dynamic fluxional behavior in the working catalyst structure. The observed dynamic fluxional behaviors give rise to motion artifacts and contrast features in the time-averaged *operando* TEM images that are similar to those discussed above. Here we observe that the Pt nanoparticles evolve from an initially well-defined, facetted morphology, with clearly visible lattice fringe contrast (Fig. 4a), to darker patches of nearly uniform contrast, now exhibiting an apparently rounded morphology, with lattice fringes that are greatly diminished in visibility (Fig. 4c). Additionally, the Ce sites on the free CeO$_2$ surface neighboring the metal-support interface become increasingly more blurred with larger catalytic turnover. Finally, in addition to a pronounced blurring at higher turnover, the top layer of Ce that neighbors the Pt/CeO$_2$ interface also displays an outward surface relaxation which grows larger with increasing activity. The outward surface relaxation is difficult to see by eye in the image but is easily measurable in intensity line profiles taken at each condition (discussed below and presented in Supplementary Fig. 15 and Supplementary Table 3).

A quantitative analysis of the *operando* ETEM images was performed in order to establish correlations between these fluxional dynamics and the catalyst's turnover frequency for CO oxidation. To describe the degree of structural dynamics taking place in the Pt nanoparticles, we quantify the Pt lattice fringe visibility (Fig. 5) through FT analysis. Windowed regions centered on the left and right Pt NPs were extracted from the time-averaged *operando* ETEM images at each condition, then filtered with a Hanning function, and finally transformed into Fourier space. The left and right Pt NP FT windows are shown, at 144 °C for example, in the left-most side of Fig. 5. Figures 5(a1–a3) and 5(b1–b3) show the modulus of the FTs taken from the right and left Pt NP, respectively, at conditions corresponding, respectively, to catalytic TOFs of 0.00, 0.80, and 1.05 CO site$^{-1}$ sec$^{-1}$.

The Pt fringe visibility clearly decreases with increasing catalytic turnover. In Fig. 5(a1), labels are given to Bragg spots corresponding to various low-index Pt fringes. The Pt lattice fringe visibility can be quantified by summing the magnitude of the Bragg spots corresponding to Pt (here we sum the FT modulus within the regions defined by the dashed yellow circles as shown in Fig. 5(a1); the image intensities were normalized prior to summation). We call this quantity the Pt FT component strength and plot it for the left and right Pt NP as a function of catalytic turnover frequency in Fig. 5c. The component strength of the left (green triangles) and right (blue squares) Pt NP are similar at each temperature. As the turnover frequency for CO oxidation increases, the visibility of fringes in either NP decreases. The component strengths of the left and right NP are 40.6 and 46.3, respectively, at 144 °C when the TOF is 0 CO site$^{-1}$ sec$^{-1}$. At 275 °C, the TOF increases to 0.80 CO site$^{-1}$ sec$^{-1}$ and the left and right Pt NP component energies drop to 30.6 and 33.4, respectively. At 285 °C, where the TOF has increased to 1.05 CO site$^{-1}$ sec$^{-1}$, the left and right Pt NP component energies decrease again to 25.1 and 28.6, respectively. As seen in the time-averaged *operando* TEM images (Fig. 3c) and in the associated FTs (Fig. 5(a3) and 5(b3)), at this condition, the lattice fringes are diffuse in either Pt NP and Bragg spots are significantly weaker.

Next we focus on the fluxional behavior and structural dynamics occurring on the free (111) CeO$_2$ surface that connects the two Pt particles. Enhanced views of the connecting terrace are given in Fig. 6 for TOFs of (a1) 0.00 CO site$^{-1}$ sec$^{-1}$, (a2) 0.80

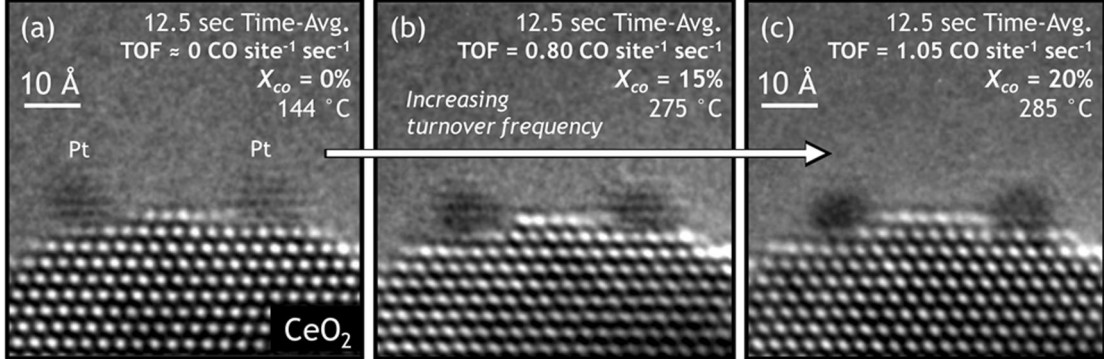

**Fig. 4 *Operando* TEM images showing dynamic structural evolution of the Pt/CeO$_2$ catalyst at varying levels of activity for CO oxidation. a** 12.5 s time-averaged image acquired at 144 °C, where the turnover frequency (TOF) was measured to be 0 CO site$^{-1}$ sec$^{-1}$. **b** 12.5 s time-averaged image acquired at 275 °C, corresponding to a TOF of 0.80 CO site$^{-1}$ sec$^{-1}$. **c** 12.5 s time-averaged image acquired at 285 °C, corresponding to a TOF of 1.05 CO site$^{-1}$ sec$^{-1}$. The corresponding temperatures and CO conversions are stated in the respective figures.

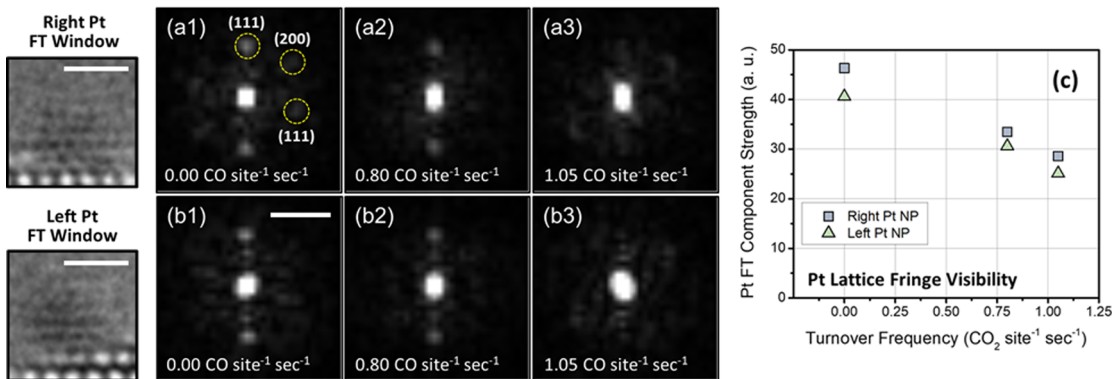

**Fig. 5 Quantification of Pt nanoparticle (NP) fringe visibility and correlation with turnover frequency for CO oxidation.** The fringe visibility is determined through Fourier transform (FT) analysis of windowed regions centered on the Pt NPs shown at far left. Scale bars in images correspond to 1 nm. **a1–a3** The modulus of the FTs after applying a Hanning filter to the windowed images of the right Pt NP at conditions corresponding to TOFs of 0.00, 0.80, and 1.05 CO site$^{-1}$ sec$^{-1}$, respectively. **b1–b3** The modulus of the FTs taken from the left Pt NP are shown at the same respective conditions. The scale bar in (**b1**) corresponds to 5.0 nm$^{-1}$. **c** The Pt fringe visibility is quantified and plotted as a function of catalytic turnover frequency.

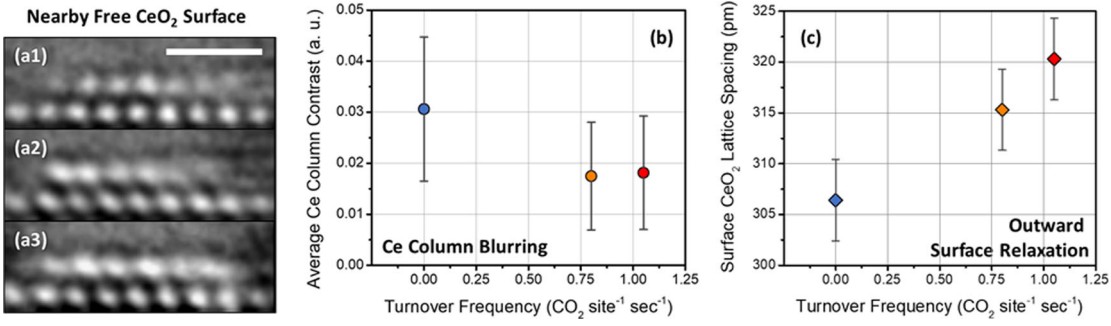

**Fig. 6 Quantification of fluxional behavior on the free CeO$_2$ (111) surface near the Pt/CeO$_2$ interface and correlation with TOF for CO oxidation. a1–a3** Magnified TEM images extracted from Fig. 4 of the nearby free CeO$_2$ (111) surface are shown for TOFs of 0.00, 0.80, and 1.05 CO site$^{-1}$ sec$^{-1}$, respectively. The scale bar in (a1) corresponds to 1 nm. **b** The average surface Ce atomic column contrast is plotted against catalytic TOF; error bars are given as the standard deviation relative to the mean measurement. **c** The separation distance between the surface and subsurface CeO$_2$ (111) lattice planes is correlated with catalytic TOF, revealing an outward surface relaxation that grows larger with increasing activity. Error bars are derived from the standard deviation about the mean lattice spacing measured in the subsurface and bulk CeO$_2$.

CO site$^{-1}$ sec$^{-1}$, and (a3) 1.05 CO site$^{-1}$ sec$^{-1}$. The surface Ce sites neighboring the metal-support interface become increasingly blurred with higher catalytic turnover. As can be seen in Fig. 6(a1), the free CeO$_2$ surface sites neighboring the Pt/CeO$_2$ interface have already shown a visible degree of blurred intensity at a TOF of 0.00 CO site$^{-1}$ sec$^{-1}$. However, when the TOF

increases to 1.05 CO site$^{-1}$ sec$^{-1}$, the blurring of the surface Ce becomes further pronounced to the point that the top layer appears as a nearly continuous band of intensity running parallel to the (111) CeO$_2$ support surface (Fig. 6(a3)). A similar blurring of the free surface Ce sites located at the opposite sides of the Pt nanoparticles can be observed in Fig. 4c. The blurring does not

appear within the $CeO_2$ bulk, indicating that the contrast change in the image is a consequence of fluxional changes in the $CeO_2$ surface near the metal-support interface.

The degree of blurring can be quantified and correlated with TOF by computing the average free surface Ce atomic column contrast at each condition. Here, the atomic column contrast is calculated by analyzing 100 pm wide intensity line profiles taken over the region that contains the surface Ce cation columns. A description of the approach along with the intensity line profiles used in the analysis is given in Supplementary Fig. 12. The average surface Ce atomic column contrast is plotted as a function of catalytic turnover frequency in Fig. 6b; error bars are given as the standard deviation relative to the mean measurement.

The contrast measurements plotted in Fig. 6b show unambiguously that the fluxional behavior giving rise to the Ce column blurring becomes more pronounced at higher catalytic turnover. At 144 °C when the TOF is 0 CO site$^{-1}$ sec$^{-1}$, the average surface Ce contrast is 0.031. At 275 °C, the TOF has increased to 0.80 CO site$^{-1}$ sec$^{-1}$, and the average contrast has decreased to a value of 0.017. At 285 °C the TOF has again increased to 1.05 CO site$^{-1}$ sec$^{-1}$, although the average measured contrast remains more or less the same at a value of 0.018. Upon further investigation, it became clear that the contrast measurement at 275 °C was biased toward lower values due to the fact that the support particle had tilted, leading to an enhanced streaking in the direction of the measurement. An examination of intensity line profiles taken from the bulk of the $CeO_2$ shows an analogous reduction in contrast (Supplementary Fig. 13b). As the catalyst was heated up to 285 °C, though, the crystal tilted back to roughly the same orientation, evidenced by the recovered contrast from columns in the bulk of the nanoparticle as seen in Supplementary Fig. 13b. Since the $CeO_2$ support has returned to roughly the same [110] zone axis orientation at 285 °C, comparing the quantified contrast of the columns at 144 °C vs 285 °C can provide insight into the evolution of blurring that occurs at individual surface Ce sites near the metal-support interface. Supplementary Fig. 14 presents the quantified contrast of individual surface Ce columns at a TOF of 1.05 CO site$^{-1}$ sec$^{-1}$ plotted against the quantified contrast of the same column when the TOF was measured to be 0 CO site$^{-1}$ sec$^{-1}$. The plot demonstrates that while on average the contrast decreases with increasing turnover, one column (located in the middle of the surface row) actually shows a slight increase in contrast, which indicates that redistributions of cation column occupancy along the surface are occurring as well as the dynamic behavior, which gives rise to the observed blurring.

In addition to a pronounced blurring at higher turnover, the top layer of Ce that neighbors the $Pt/CeO_2$ interface also displays an outward surface relaxation which grows larger with increasing activity. The outward $CeO_2$ (111) surface relaxation was quantified by measuring the lattice plane separation distance from intensity line profiles taken from the interior of the $CeO_2$ support toward the catalyst surface. A description of the approach along with the intensity line profiles used in the analysis is provided in Supplementary Fig. 15. The separation distance measured between the surface and subsurface $CeO_2$ (111) lattice planes is plotted as a function of catalytic turnover frequency in Fig. 6c. The error bars are derived from the standard deviation about the mean lattice spacings measured in the subsurface and bulk at that condition (here, we briefly note that the average of the subsurface and bulk lattice plane separation distances was measured to be 310 ± 4 pm, which agrees with the accepted $CeO_2$ (111) Miller plane spacing of 312 pm).

Examining the surface lattice spacing plotted against catalytic turnover frequency illustrates the emergence of an outward surface relaxation that grows linearly with increasing activity. At 144 °C where the TOF is 0 CO site$^{-1}$ sec$^{-1}$, the surface $CeO_2$ (111) lattice

plane distance is 306 ± 4 pm. At 275 °C, where the TOF has increased to 0.80 CO site$^{-1}$ sec$^{-1}$, the surface $CeO_2$ (111) lattice plane has relaxed outward by 9 pm to a value of 315 ± 3 pm. At 285 °C, the TOF has increased to 1.05 CO site$^{-1}$ sec$^{-1}$, and the surface $CeO_2$ (111) lattice plane has relaxed by another 5 pm to a value of 320 ± 5 pm. The surface lattice plane separation was also measured in an inert atmosphere of $N_2$ at room temperature and at working temperature; no such expansion was detected upon heating the catalyst up to 300 °C, as shown in Supplementary Fig. 11.

Considering the substantial dynamic structural behavior as well as the outward $CeO_2$ support surface relaxation observed to occur during catalysis, one may suspect significant local lattice strain to be present. We have measured the strain in the $CeO_2$ cation sublattice by fitting 2D elliptical Gaussians to the Ce columns in the time-averaged image[45] in order to determine their position; then we have calculated the difference in each columns' position with respect to the bulk terminated lattice, in a standard way. We call this quantity the average static strain since it is measured from the time-averaged images, which are the sum of image signals recorded over a total time period of 12.5 s. The in-plane component of the average static strain parallel to the $CeO_2$ (111) surface is plotted in Fig. 7 for TOFs of (a) 0.00 CO site$^{-1}$ sec$^{-1}$ and (b) 1.05 CO site$^{-1}$ sec$^{-1}$ (strain analysis for data corresponding a TOF of 0.85 CO site$^{-1}$ sec$^{-1}$ was not performed due to the aforementioned crystal tilt at this condition). In the figure, the fitted positions of Ce atomic columns are represented by circles. The in-plane component of the average static strain shows rapid spatial variation between tensile and compressive across the free $CeO_2$ surface terrace, with a magnitude in the range of 15–20%.

While the static strain at the surface is large, it does not appear to show any clear correlation with the TOF. We believe this is because the average static strain may not give an accurate representation of the dynamic bond distortion that may be present due to fluxional behavior at any given time. For example, if during an image exposure time, $t$, the cation spends a time $t/2$ at distance $+d$ away from its bulk terminated position and a time $t/2$ at a distance $-d$, then the average cation displacement over the observation time would be zero. To address this deficiency, we define a fluxional strain quantity that is proportional to the width of the atomic column in the time-averaged image. The fluxional strain, $\varepsilon_{flux}$, can then be taken as the difference in the standard deviation of the Gaussian fitted to the particular Ce cation column of interest, $\sigma_{local}$, and the average of the bulk, $\sigma_{bulk}$, with respect to the ideal in-plane $CeO_2$ (111) Ce cation spacing of 294 pm, $d_{ideal}$, and is defined as follows:

$$\varepsilon_{flux} = \frac{(\sigma_{local} - \sigma_{bulk})}{d_{ideal}} \qquad (2)$$

The local magnitude of the fluxional strain is visually represented in Fig. 7 by the size of the circles, which are proportional in dimension to the standard deviation of the Gaussian fitted to the Ce atomic columns. Note that the fluxional strain of the atomic columns in the bulk of the $CeO_2$ does not vary significantly with increasing turnover frequency. The columns at the surface, however, grow wider with increasing turnover, as can be seen by examining the circles drawn at the top of the figures in the arrowed locations, which correspond to the atomic columns located at the free $CeO_2$ (111) surface terrace. The average fluxional strain across the surface terrace is around 19% at 144 °C when the conversion is zero. However, as the conversion increases, the fluxional strain also increases and is around 33% at 285 °C when the conversion is 15%. This is a large degree of strain, and the resultant high surface energy will make the catalyst surface extremely reactive.

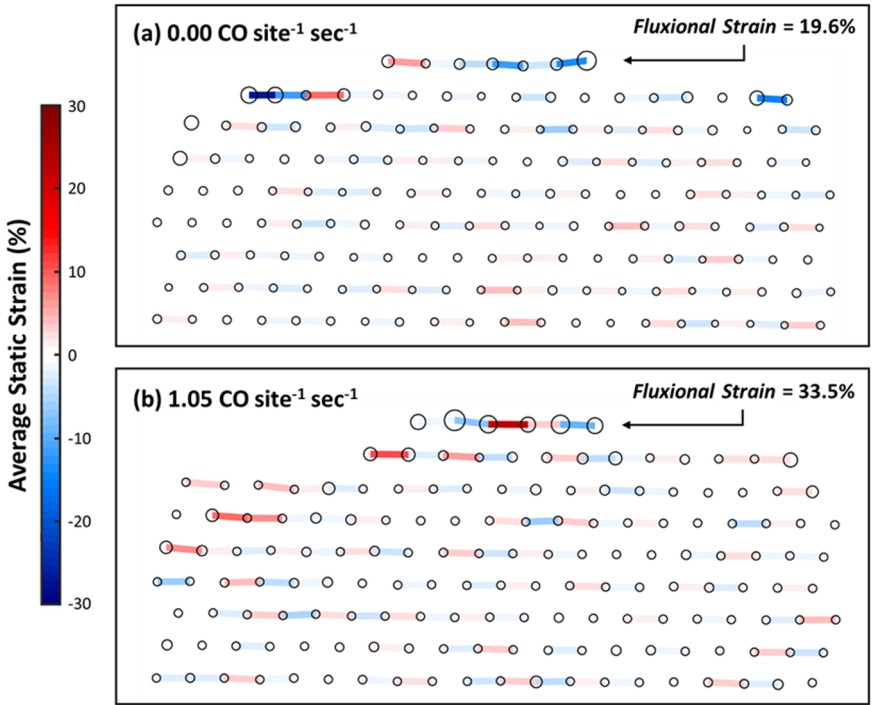

**Fig. 7 Average static and fluxional strain in CeO$_2$ support. a** Static and fluxional strain of Ce cation sublattice at a TOF of 0.00 CO site$^{-1}$ sec$^{-1}$. **b** Static and fluxional strain in the Ce cation sublattice at a TOF of 1.05 CO site$^{-1}$ sec$^{-1}$. The circles signify the Ce atomic column position determined by Gaussian peak fitting and are drawn proportional in size to the standard deviation of the fitted Gaussian. The strain between sites is represented with a colored bar; red and blue correspond to tensile and compressive strain, respectively. The free CeO$_2$ (111) surface terrace between the supported Pt nanoparticles has been marked with a black arrow. (The Pt particles are not shown).

## Discussion

The observations and measurements clearly show significant dynamic changes in the metal particles, the metal/ceramic interface, and the nearby support surfaces under reaction conditions and during catalysis. The rate of structural dynamics at or near the three-phase boundary correlates with the TOF and thus provides insights into the atomic-level materials processes taking place during a Mars van Krevelen catalytic reaction. Interestingly, the structural changes are observed to some degree even when the TOF is zero. However, as we will discuss below, it would be an oversimplification to conclude that these are spectator processes. The fluxionality is caused by making and breaking chemical bonds on the surface of the catalyst, i.e. it is associated with surface chemistry and not just the rate-limiting step for complete oxidation of CO. The detection of CO$_2$ in the gas phase implies that the rate-limiting step has been completed, but there are other intermediate steps in the reaction pathway which will have lower activation energies and will proceed at lower temperatures. In the case of CO oxidation via a Mars-van Krevelen mechanism, there are many steps in the reaction pathway including:

- Chemisorption of CO onto Pt nanoparticle
- Abstraction of interfacial O from the CeO$_2$/Pt perimeter sites by chemisorbed CO species
- Reduction of molecular O$_2$ to backfill oxygen vacancies

The first two steps can take place even at room temperature. For example, we recently showed that, even on pure CeO$_2$ at room temperature, oxygen vacancies are created and annihilated at special surface sites such as step edges, strained terraces, nanofacets, etc.[25]. This process does not involve the reduction of molecular oxygen (which is not activated on the bare CeO$_2$ surface at room temperature), but rather the abstraction of oxygen from surface lattice sites. In the presence of Pt nanoparticles and

in CO, this process becomes considerably more active. This was demonstrated in our recent conference publication[41] where we showed that, at room temperature, Pt nanoparticles can show intense fluxional behavior when CO is introduced but significantly less in N$_2$ gas. The destabilization of the Pt nanoparticle is a consequence of the abstraction of oxygen from the metal-support interface sites, as discussed below.

The catalyst changes form as the TOF increases due to the substantial increases in fluxional behavior. The fact that the changes become more pronounced when catalysis is detected suggests that they are integral to the functioning of the Mars van Krevelen mechanism in this case. The various dynamic processes are interconnected, but it is helpful to initially discuss them in detail separately.

The most obvious change observed in our experiments is the emergence of structural dynamics taking place in the Pt particle under reaction conditions. Significant fluxional behavior of the Pt nanoparticles takes place even at 144 °C, which is below the light-off temperature for the catalyst. We also observed similar behavior in recent in situ TEM[26] on the water gas shift reaction at 200 °C and we have even observed significant fluxional behavior in Pt particles supported on CeO$_2$ at room temperature in atmospheres containing CO[46] (but not in N$_2$, for example see Supplementary Figures 9 and 10). It is clear that this behavior is not directly associated with the rate-limiting step for CO oxidation, but one may conclude that the equilibrium shape of 1–2 nm Pt nanoparticles supported on CeO$_2$ is not well-defined in a CO atmosphere over a wide range of temperatures.

Under equilibrium conditions, a supported nanoparticle will adopt the so-called Winterbottom shape[47] which minimizes the sum of the surface and interfacial energies. For an FCC metal like Pt, the (111) surface has the lowest energy in vacuum[48] and the coherent interfaces associated with strong bonding (e.g., the one shown in Fig. 1)

are also low energy. In this case, the strong interfacial bond between Pt and $CeO_2$ is associated with bridging oxygen giving rise to Pt–O–Ce linkages[12–16]. For these metallic systems, the Winterbottom shape often gives rise to a truncated Wulff shape (the equilibrium shape of an unsupported particle). In the presence of gas adsorbates, the surface energies may change, causing the equilibrium particle shape to change[49,50]. As is well known, CO binds very strongly to Pt with average chemisorption energy of about 1.3–1.5 eV[9,51]. It is important to recognize that the formation of this strong Pt-CO bond will also weaken the Pt bond with its nearest neighbors. Under reaction conditions, most surface Pt atoms will be bonded to CO[9,51], and so there will be a substantial weakening of the bond between the surface and the subsurface Pt atoms.

Examination of the Ce cations at the Pt perimeter sites in Fig. 4a shows that oxygen vacancy creation and annihilation is taking place due to interaction with CO adsorbed on the Pt. For 2 nm Pt particles, assuming a simplified hemispherical cubocta-hedral shape, approximately 35% of the Pt atoms are located at the Pt/$CeO_2$ interface and about 40% of those atoms occupy the perimeter sites. Oxygen vacancy creation at these perimeter sites removes bridging oxygen and weakens the bonding between the Pt nanoparticle and support. The fluxional behavior in the surface Ce cations suggests that there is constant competition between vacancy annihilation, which repairs the Pt-O-Ce bond, and vacancy creation, which is due to reaction with CO. We can get additional temporal insights by considering the image exposure time. Over a 0.5 s exposure at 144 °C (see Fig. 2b-e), the significant blurring of the perimeter Ce cations indicates that many vacancies are created and annihilated due to interactions with CO.

At this temperature, no $CO_2$ is detected, so the carbonaceous intermediates may remain bound to the catalyst's surface (indeed, several authors report that $CO_2$ adsorbed on *reduced* $CeO_{2-x}$ binds with adsorption energy in excess of 1 eV, as discussed by D. R. Mullins[46]). Ionic oxygen transport is also small at 144 °C, so there is a limited availability of lattice oxygen to backfill the surface and interfacial vacancies. Thus, much of the vacancy creation/annihilation will be associated with forward and reverse reactions between surface lattice O and CO, i.e. this intermediate step may be highly reversible. The continuous disruption of the Pt-O-Ce interfacial bond and the changes in adsorbate configuration on the Pt surface disrupt the Winterbottom shape and leads to the observed dynamic behavior.

As carbon species spill over onto the adjacent $CeO_2$ surface, additional CO will adsorb onto the vacant surface sites on Pt. This dynamic bonding and debonding of adsorbates on the metal surface lead to distortions and strains, driving surface diffusion. In recent work, we correlated in situ TEM and in situ synchrotron observations. The synchrotron analysis showed that, even when rapid fluxional behavior is occurring, the average Pt coordination and bond lengths are consistent with the Pt FCC crystal structure[26]. This shows that the nanoparticle is not forming an amorphous structure, but rather it is transforming rapidly between metastable configurations. At present, our temporal resolution and signal-to-noise preclude a detailed identification of the structure of these metastable phases. However, the analysis of the visibility of the Bragg beams in the diffractograms shows that the rate of reconfiguration increases with conversion. This is consistent with the more frequent formation and breaking of surface and interfacial chemical bonds associated with high catalytic TOF.

As the temperature and conversion increase, there is a substantial increase in the fluxional behavior on the $CeO_2$ surface. While the highest vacancy creation/annihilation activity takes place at the perimeter sites, the fluxional behavior extends over the entire 20 Å nanofacet between the two Pt particles shown in

Fig. 6c. The outward relaxation of the surface layer is also consistent with a high concentration of oxygen vacancies along the surface. No lattice expansion was detected in the subsurface layer, and by the third layer down, no evidence is seen for any Ce cation blurring. This localized surface fluxional behavior indicates that most of the oxygen vacancy creation and annihilation as well as oxygen diffusion takes place on the top layer. Negligible oxygen transport from the bulk should take place at these temperatures since the activation energy of undoped $CeO_2$ has been measured to lie in the range 0.9–2.3 eV[52,53]. At conditions of detectable conversion, this implies that at least some of the oxygen vacancies must be annihilated via reduction of molecular oxygen in order to replenish the supply of oxygen required for steady-state oxidation of CO.

One may ask where is the likely site for molecular oxygen reduction? It could take place at the oxygen vacancies created at the perimeter sites after $CO_2$ desorbs. Alternatively, it could take place on the nearby free $CeO_2$ terrace and then diffuse to the perimeter sites. As shown in Fig. 7, the average static surface strain is high on the free $CeO_2$ surface, and the fluxional strain on the surface terrace reaches around 30% when the conversion is 15%. Without sufficiently time-resolved information on the Ce atomic columns, we hesitate to comment specifically on the character of the behavior giving rise to the observed dynamic strain, e.g., whether it is due to positional displacements, surface atomic diffusion, or extended defects such as shear planes – nevertheless, this is a very large strain, and the associated high surface energy will make the terrace more reactive making direct oxygen reduction feasible. At present we are not able to differentiate definitively between either perimeter or terrace mechanism, though it may be worth speculating on the likelihood of each. We do not have atomic-resolution surface spectroscopy to determine the nature of the carbon species created as oxygen vacancies are created and annihilated. However, carbonates are known to form during CO oxidation on $CeO_2$[46,54], so a possible reversible reaction that may occur at the interface is:

$$2(Pt--O--Ce) + CO_{ad} \leftrightarrow 2(Pt----Ce) + ((CO_3)^{-2})ad$$

The perimeter environment may be rather crowded with a high concentration of CO on the Pt sites and a high concentration of carbonates on the $CeO_2$ sites. Backfilling of the interfacial oxygen vacancies by direct adsorption of molecular oxygen may be subject to significant steric hindrance. It may be easier to annihilate the interfacial oxygen vacancy by diffusion of lattice surface oxygen along the nearby $CeO_2$. During catalysis, oxygen reduction almost certainly takes place on the highly reduced nearby $CeO_2$ terrace and that lattice oxygen migrates to the perimeter sites for reaction with CO.

These experiments provide an atomic-level view of the structural dynamics associated with the three-phase boundary of the catalyst during Mars-van Krevelen oxidation. The localized structural changes that are observed are directly associated with catalytic functionality. Many of the steps for converting reactants into products involve forming and breaking chemical bonds with the atoms forming the catalyst surface. This bonding and debonding not only changes the adsorbates but may locally destabilize the catalyst surface and interface structures, resulting in large local surface strains, surface atom migration, and the creation and annihilation of point defects. The particular fluxional behaviors reported here are specific to the Pt/$CeO_2$ catalytic system and CO oxidation. However, breaking and forming chemical bonds is an essential functionality for all heterogeneous catalysts, and it can only occur if the catalytic surface undergoes substantial structural dynamics. While the electron transfer occurring at an active site happens on the femto second time scale, the subsequent nuclear rearrangements may alter or completely destroy the active site due to the thermodynamic and

kinetic factors driving structural change. The question of how to explore structure-reactivity relations then becomes more complex: in at least some nanoparticle catalysts, it may be more appropriate to carefully consider the atomic-level structural dynamics that may be integral to the catalytic cycle and reactivity. The importance of adsorbate-induced structural changes has long been recognized in the surface science community[18]. Now, however, the picometer precision of advanced *operando* TEM can reveal the local, atomic-level details of these structural dynamics for actively working technical catalysts.

In summary, we have employed aberration-corrected *operando* TEM to visualize the atomic-scale dynamic structural (i.e., fluxional) behavior occurring at and near Pt/CeO$_2$ metal-support interfaces caused by oxygen transfer during CO oxidation. Finite element modeling is performed to develop a reaction rate analysis that allows the catalyst's turnover frequency to be quantitatively correlated with different forms of structural dynamics processes. This analysis reveals a direct connection between activity and certain forms of fluxional behavior that are uniquely relevant to the catalytic reaction (not temperature). In this case, we show that the catalytically-relevant fluxional dynamics are associated with an enhanced rate of oxygen vacancy creation and annihilation on the surface at and close to the Pt perimeter sites, which leads to increased fluxional strain in the cation sublattice of the CeO$_2$ support and an enhanced overall reduction in the support surface. These results unambiguously demonstrate that there are dynamic and ongoing structural reconfigurations taking place around the metal-support interface which are correlated directly with catalysis. Overall, the results implicate the interfacial Pt-O-Ce bonds anchoring the Pt to the support as being involved also in the catalytically-driven oxygen transfer process, and they suggest that molecular oxygen reduction takes place on the highly reduced nearby CeO$_2$ surface before migrating to the metal-support interfacial perimeter for reaction with CO. This study highlights the importance of characterizing the structural dynamics that take place during catalysis in order to elucidate the relationship between a catalyst's structure and its functionality.

## Methods

**Catalyst preparation and structural/activity characterization**. The catalyst, which consists of Pt nanoparticles supported on nanoparticles of CeO$_2$, was produced by standard hydrothermal[55] and incipient wetness impregnation methods, which are described in detail in the Supplementary Information. A high weight loading (17 wt.%) of Pt metal was desired to reduce the amount of time spent searching for Pt nanoparticles close to a zone-axis orientation during in situ and *operando* TEM experiments. The as-prepared catalyst powder was thermally processed in a flowing stream of 5% H$_2$/Ar for 2 h at 400 °C prior to any structural or catalytic activity characterization.

Structural characterization was performed using X-ray diffraction (XRD) and aberration-corrected (scanning) TEM ((S)TEM). XRD patterns were collected on a Bruker D-5000 with a Cu Kα source (λ = 0.15406 nm). HRTEM images were collected on a Thermo Fisher Titan operated at 300 kV, with the lens system's 3$^{rd}$-order spherical aberration coefficient (-Cs) tuned to approximately −13 μm to yield increased white-column phase contrast[56]. The STEM (a probe-corrected JEOL ARM200F) was operated at 200 kV and high angle annular dark-field (HAADF) images were collected to determine the Pt nanoparticle size distribution, which allowed for an estimate to be made of the number of Pt atoms located at the interfacial perimeter. A derivation of this estimation is given in the Supplementary Information.

Catalytic activities for CO oxidation were evaluated in a packed bed plug-flow reactor. Details on the experimental conditions are given in the Supplementary Information. Plug flow reactor experiments were performed in a RIG-150 microreactor from In Situ Research Instruments (ISRI). Effluent gas compositions were measured with a Varian 3900 gas chromatograph (GC) equipped with a thermal conductivity detector (TCD).

**Atomic-resolution operando environmental TEM**. Atomic-resolution *operando* ETEM experiments were performed on an image-corrected Thermo Fisher Titan ETEM at 300 kV equipped with a Gatan Imaging Filter for electron energy-loss spectroscopy (EELS). An *operando* pellet reactor approach was employed to increase the mass of catalyst in the microscope and facilitate the determination of

chemical kinetics during atomic-resolution imaging[43,57,58]. An overview of the reactor geometry is given in Supplementary Fig. 4. In this approach, a Gatan 628 furnace-style Ta hot stage is used for heating. The Ta is inactive. The furnace is loaded with a porous glass fiber pellet and an inert Ta metal TEM grid. Catalyst particles are dispersed on the TEM grid and loaded into the glass fiber pellet through aqueous drop-casting. The mass of the catalyst in the pellet was measured with a microbalance and determined to be 180 ± 5 μg. When heated in reactant gases, the large mass of catalyst on the pellet facilitates the production of detectable conversions. The fiber pellet is annular in shape, having a 1 mm wide hole at its center, which permits the electron beam to pass through unobstructed for spectroscopy and imaging. The TEM grid is visible in the region containing the hole, and it is conducting, so it provides a stable platform for atomic-resolution imaging. The gas composition in the cell can be quantified with EELS [42].

After inserting the catalyst-loaded furnace reactor into the ETEM, an in situ reduction step was performed in 1 Torr of H$_2$ at 400 °C for 2 h. The stage was then cooled to 120 °C and the column was briefly evacuated. As is common for CO oxidation, an oxygen-rich reactant environment was used; in this experiment, 0.24 Torr of CO and 0.33 Torr of O$_2$ was admitted into the cell and the reactor was heated up to 300 °C. The pressure of gases used in the ETEM should not significantly impact the kinetic catalytic mechanism given the well known 0$^{th}$-order dependence of the rate on the CO and O$_2$ partial pressures[6,59]. Carrier gases were not used to increase the relative partial pressures of the reactants and to achieve high signal-to-background levels for EELS gas composition measurements.

*Operando* TEM images were acquired in the aforementioned negative Cs imaging mode with an incident electron dose rate of ~1 × 10$^3$ e$^-$ Å$^{-2}$ s$^{-1}$ and an exposure time of 0.5 s on a Gatan Ultrascan 1000 CCD detector. Previous ETEM work[60] investigating the effect of the incident electron beam dose rate on the structure and chemistry of CeO$_2$ surfaces has demonstrated that electron beam effects are not significant for imaging current densities below 6 × 10$^3$ e$^-$ Å$^{-2}$ s$^{-1}$. Additionally, Sinclair et al. (2017) report that no observable structural or chemical changes occur for this dose when low pressures of ambient oxygen are present[60]. Here, oxygen is present, an electron flux below the reported critical threshold is used, and the dose rate is maintained below this level throughout the experiment. Furthermore, work by Lawrence et al. (2021) shows that, for the imaging conditions employed in the present work, the rate of electron beam-induced radiolytic or knock-on events becomes negligible compared to that of thermally-induced events (i.e., those expected to regulate catalytic processes) as the temperature of the CeO$_2$ rises above 150 °C, which is what is explored here[25].

The use of a low electron dose rate resulted in a poor signal-to-noise ratio (SNR) in the 0.5 s exposure images that were acquired (for example, see one of the frames shown below in Figs. 2b–2e). Utilizing a longer image acquisition time was impractical due to instabilities associated with drift in the furnace hot stage required for *operando* TEM. Thus, a drift-corrected, time-averaged image method was implemented to increase the SNR available for local structural analysis under *operando* conditions. A schematic and description of the approach is provided in Supplementary Fig. 6.

In situ EELS was implemented to track the gas composition[42,43,44]. All spectra were acquired in diffraction mode using an entrance aperture of 2 mm, a camera length of 245 mm, a dispersion of 0.05 eV pixel$^{-1}$, and an acquisition time of 5 s. Given the 1:1 stoichiometry of C in the CO oxidation reaction, the in situ CO conversion was calculated to be the fractional amount of CO$_2$ to CO and CO$_2$ in the cell.

**Finite element simulation of operando ETEM reactor**. Recently, we have developed a finite element model of the ETEM reactor, allowing for an investigation into how the reactant conversion measured along the EELS line can be used to derive the steady-state reaction rate for catalyst supported on the Ta TEM grid[34]. Here we have employed the model to simulate the *operando* ETEM reactor under conditions matching the present experiment in order to establish a framework for chemical kinetic analysis. The catalytic reaction was modeled as 0th order, with kinetic parameters describing the activity of the catalyst taken from the Arrhenius analysis from the plug flow reactor data (Supplementary Fig. 3). The spatial distribution of catalyst mass in the pellet was modeled with an egg-shell profile. Simulations were performed in the commercial program COMSOL Multiphysics®. More information describing the simulations is given in[34] and in the SI.

**In situ ETEM in non-reactive gases at elevated temperature**. In situ ETEM experiments were conducted on the Pt/CeO$_2$ catalyst in an atmosphere of inert N$_2$ both at room temperature and at 300 °C to differentiate spectator structural dynamics associated with applied heat from those associated with catalysis. These experiments were conducted in an image-corrected Thermo Fisher Titan ETEM operated at 300 kV. TEM samples were prepared by dispersing the Pt/CeO$_2$ powder onto a windowed micro-electro-mechanical system (MEMS)-based SiN$_x$ chip, which was then loaded into a DENSsolutions Wildfire heating holder. After loading the holder into the ETEM, 5 mTorr of N$_2$ gas was introduced into the environmental cell, and the catalyst was imaged at room temperature. The specimen was then heated to 300 °C and the catalyst was imaged again. Images were acquired with an incident electron beam dose rate of 5 × 10$^3$ e$^-$ Å$^{-2}$ s$^{-1}$ using a Gatan K2 IS direct detector in the electron counting mode. The beam was blanked during the

in situ heating and whenever images were not being acquired to further minimize electron beam-induced changes.

## Data availability

The data supporting the findings of this study are available within the article and its Supplementary Information file. Other data are available from the authors upon request.

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

## Acknowledgements

The authors gratefully acknowledge funding for this research from NSF grant CBET 1604971. The authors thank Arizona State University's John M. Cowley Center for High Resolution Electron Microscopy for microscope access and use. The authors also gratefully acknowledge the use of environmental electron microscopy facilities at the National Institute of Standards and Technology in Gaithersburg, MD, and in particular would like to acknowledge the helpfulness and hospitality of Dr. Wei-Chang David Yang, Dr. Canhui Wang, and Dr. Renu Sharma. Additionally, the authors are grateful to Mr. Piyush Haluai for assistance with measuring the strain of Ce atomic columns.

## Author contributions

JLV performed the experiments and analysis. JLV wrote the manuscript with input from PAC. PAC conceived of the research, supervised all aspects of the project, and edited the manuscript.

## Competing interests

The authors declare no competing interests.
