## [Peer Review File · Nature Communications]

Atomic Level Fluxional Behavior and Activity of
CeO₂-supported Pt Catalysts for CO OxidationREVIEWER COMMENTS

Reviewer #1 (Remarks to the Author):

This is a very interesting and relevant study on the role of the dynamical behavior of a heterogeneous catalyst under operating conditions. The work reports an atomic-scale analysis of the meta-stability and fluxional behavior of a Pt nanocluster supported on CeO₂ during a classical Mars Van Krevelen CO oxidation reaction. Using aberration-corrected operando electron microscopy the authors visualize the structural dynamics occurring at the Pt/CeO₂ interface and use finite element simulations to develop a reaction rate analysis. In this way they are able to correlate the observed structural modifications with the catalyst's turnover frequency. The important and novel aspect is that not only the fluxionality of the supported Pt particle is detected, but also the changes in the supporting oxide occurring both at the metal/oxide interface and at the ceria surface between the supported Pt particles. The results are of excellent quality, and the paper is very well written and pleasant to read. I think this paper should be published in Nat. Communications in essentially the present form.

The only aspect that I found a bit surprising, but the authors seem to find it surprising as well, is the large fluxional strain of the ceria surface that reaches about 30% under certain conditions. This is an extremely large strain, which under normal circumstances could be attributed to atomic diffusion under the top ceria layer or encapsulation of some metal atoms. I know that this point is specifically addressed in the paper, but nevertheless a more detailed atomistic description of this large strain would be useful.

Reviewer #2 (Remarks to the Author):

The manuscript by Vincent and Crozier reports results of operando electron microscopy of CO oxidation on Pt nanoparticles supported on CeO₂. The authors correlate dynamic restructuring of the catalyst and localized turnover rates based on a finite elements modeling in three kinetic regimes, associated with no, medium, and high TOFs. They identify an increased fluxionality of the Pt particle as well as the CeO₂ support in contact with the particle, when the local TOF is increased. As the image acquisition time is within the order of the TOF the fluxionality is captured by a blurring of the relevant species. The authors derive zones of surface strain in the support based on a procedure previously communicated. This identifies the anchoring of the catalyst particle as zone, where bonds are broken and formed. Overall, the manuscript reads quite well and presents interesting research.

To this reviewer it remained unclear, however, what the truly novel insight is that warrants publication in one of the highest impact journals. The authors themselves have reported a fluxional behavior of catalyst nanoparticles, e.g., Li et al., 2021. The modeling approach was also presented recently, Vincent & Crozier 2020. A similar approach was also applied to Au/CeO₂, reporting restructuring of the gold-ceria interfacial perimeters. This leaves more the impression of a short review of insights applied to the topic of CO oxidation by the group, than a clear novelty and breakthrough rewarding a high impact publication.

Here I would like to stress yet again that I found the manuscript very interesting to read, also providing context (although shortening the text would help). Therefore, my recommendation would be to either streamline this work to clearly present the novelty or even better present this as a research account of the group.

Major issues:

1) How can one explain the particle fluxionality in the low activity state? The particles should be covered by CO, but how does this lead to changes in coverage/surface energy that could induce

fluxionality ? As most of the Pt are inactive under such conditions (low CO conversion), can beam effects really be excluded ?

2) If yes, what is the effect of reconstructions as those described by G. Ertl?

3) In addition to the experiments under constant CO/O₂ feed at increasing temperature, one could also change from inactive to active state by changing the CO:O₂ ratio. Were such isothermal experiments (ruling out any temperature-induced fluxionality) also carried out ?

4) With TOFs on the order of 1 per site-1 s⁻¹ it seems that the 12.5 s time averages of Fig 4 rather capture the long-term fate of the catalyst, than the fluxionality connected to individual turnovers. What aspect of the fluxionality is truly periodic in the sense of turnovers?

5) How reliable is the finite elements prediction of turnovers given the fluxionality of the particles, that would heavily influence the local reaction rates?

Bold lettering = suggested changes by the reviewer

Green lettering = response to comment

Red lettering = changes/revisions that were implemented in the revised manuscript

Note: all references to page numbers are made with respect to the document presented in *All Markup* view.

Reviewer(s) Comments to Author:

Reviewer 1:

This is a very interesting and relevant study on the role of the dynamical behavior of a heterogeneous catalyst under operating conditions. The work reports an atomic-scale analysis of the meta-stability and fluxional behavior of a Pt nanocluster supported on CeO₂ during a classical Mars Van Krevelen CO oxidation reaction. Using aberration-corrected operando electron microscopy the authors visualize the structural dynamics occurring at the Pt/CeO₂ interface and use finite element simulations to develop a reaction rate analysis. In this way they are able to correlate the observed structural modifications with the catalyst's turnover frequency. The important and novel aspect is that not only the fluxionality of the supported Pt particle is detected, but also the changes in the supporting oxide occurring both at the metal/oxide interface and at the ceria surface between the supported Pt particles. The results are of excellent quality, and the paper is very well written and pleasant to read. I think this paper should be published in Nat. Communications in essentially the present form.

We are grateful to the reviewer for these comments.

1. The only aspect that I found a bit surprising, but the authors seem to find it surprising as well, is the large fluxional strain of the ceria surface that reaches about 30% under certain conditions. This is an extremely large strain, which under normal circumstances could be attributed to atomic diffusion under the top ceria layer or encapsulation of some metal atoms. **I know that this point is specifically address in the paper, but nevertheless a more detailed atomistic description of this large strain would be useful.**

Response: We thank the reviewer for their helpful suggestion. The reviewer is correct to note that we were at the time of writing somewhat surprised to discover the large fluxional strain on the ceria surface during catalysis. In this case, for the in-plane component of the CeO₂ (111) lattice plane, the ideal lateral Ce cation distance is 294 pm. A fluxional strain of 30% thus corresponds to lateral displacements of ~100 pm from the equilibrium column position. We have recently employed time-resolved in situ aberration-corrected TEM with a fast and efficient direct electron detector to more precisely characterize the dynamic cation behavior on CeO₂ nanoparticle surfaces. The results (reported in our recent conference publication, Tan, M., Vincent, J. L., and Crozier, P. A. (2021), *Microscopy and Microanalysis*, 27(S1), 2244-5), demonstrate that even at room temperatures, Ce cation columns located at certain sites, e.g., corners, edges, etc., can undergo dynamic displacements of at least 70 pm over ~100 ms. Under operando conditions, where the temperature is much higher and the CeO₂ surface more reduced, it is reasonable to expect larger displacements, giving rise to temporary fluxional strains on the order of 30%.

We appreciate both the reviewer's comment and the potential for alternative interpretations of the data. Without sufficiently time-resolved information on the position and occupancy of the atomic-columns, we hesitate to comment specifically on the nature of the behavior giving rise to the observed contrast features. For example, the fluxional behavior may be simply positional displacements, or it may be associated with atomic diffusion, as the reviewer suggests; additionally, in Pt/CeO₂ under CO oxidation gas atmospheres, we have observed the formation of Ce cation shear planes in the CeO₂ support (Atomic-resolution Operando and Time-resolved In Situ TEM Imaging of Oxygen Transfer Reactions Catalyzed by CeO₂-supported Pt Nanoparticles, J. L. Vincent and P. A. Crozier, *Microscopy and Microanalysis* 26 (S2), 1694-1695), and it may be possible that some element of the blurring is associated with this structural behavior, although it seems unlikely in the present case, since the blurring appears localized to the surface near the metal-support interface. We have revised the manuscript in order to clarify the description of the fluxional strain and to comment on possible alternative interpretations, e.g., atomic diffusion.

Changes to the text:

Starting on page 24, bottom of the page, through page 25, made minor revisions to improve clarity regarding the fluxional strain measurement.

On page 25, revised the definition of the fluxional strain quantity to add context and to correct an ambiguity present in the original definition: "The fluxional strain, ϵ_{flux} , can then be taken as the difference in the standard

deviation of the Gaussian fitted to the particular Ce cation column of interest, σ_{local} , and the average of the bulk, σ_{bulk} , with respect to the ideal in-plane CeO₂ (111) Ce cation spacing of 294 pm, d_{ideal} , and is defined as follows: $\epsilon_{flux} = \frac{(\sigma_{local} - \sigma_{bulk})}{d_{ideal}}$.

On page 31, added this sentence to the discussion in order to address the possibility of alternative interpretations of the large degree of fluxional strain observed during catalysis: “Without sufficiently time-resolved information on the Ce atomic columns, we hesitate to comment specifically on the character of the behavior giving rise to the observed dynamic strain, e.g., whether it is due to positional displacements, surface atomic diffusion, or extended defects such as shear planes.”

Made minor revisions in the text of the same paragraph to accommodate the new sentence.

Reviewer 2:

The manuscript by Vincent and Crozier reports results of operando electron microscopy of CO oxidation on Pt nanoparticles supported on CeO₂. The authors correlate dynamic restructuring of the catalyst and localized turnover rates based on a finite elements modeling in three kinetic regimes, associated with no, medium, and high TOFs. They identify an increased fluxionality of the Pt particle as well as the CeO₂ support in contact with the particle, when the local TOF is increased. As the image acquisition time is within the order of the TOF the fluxionality is captured by a blurring of the relevant species. The authors derive zones of surface strain in the support based on a procedure previously communicated. This identifies the anchoring of the catalyst particle as zone, where bonds are broken and formed. Overall, the manuscript reads quite well and presents interesting research.

We thank the reviewer for their comments.

1. To this reviewer it remained unclear, however, what the truly novel insight is that warrants publication in one of the highest impact journals. The authors themselves have reported a fluxional behavior of catalyst nanoparticles, e.g., Li et al., 2021. The modeling approach was also presented recently, Vincent & Crozier 2020. A similar approach was also applied to Au/CeO₂, reporting restructuring of the gold–ceria interfacial perimeters. This leaves more the impression of a short review of insights applied to the topic of CO oxidation by the group, than a clear novelty and breakthrough rewarding a high impact publication. Here I would like to stress yet again that I found the manuscript very interesting to read, also providing context (although shortening the text would help). **Therefore, my recommendation would be to either streamline this work to clearly present the novelty or even better present this as a research account of the group.**

Response: We thank the reviewer for their comments and suggestions. With all due respect to the reviewer, there may be some confusion over in situ and operando measurements. The former involves no measurement of catalysis and is focused only on identifying structural transformations taking place as a result of gas-solid interactions. Admittedly when one performs such an experiment, one often assumes that catalysis is taking place, but the reaction conditions inside a TEM are always different from those of an ex situ reactor so this assumption is easily criticized. The problem with such an approach is that the structures observed due to gas-solid interactions may be spectator structures. With no simultaneous information on chemical kinetics, it is impossible to ascertain the relevance of the structure to catalysis. We recently demonstrated this pitfall of in situ methods in our publication on CO oxidation over Ru. (Linking Changes in Reaction Kinetics and Atomic-Level Surface Structures on a Supported Ru Catalyst for CO Oxidation, B. K. Miller and P. A. Crozier, ACS Catalysis 2021 Vol. 11 Pages 1456–1463). RuO₂ is clearly observed to form under reaction conditions, but it is not the active form of the catalyst. So, observing a structure forming under reaction conditions is not enough to establish a link to catalytic functionality.

It is because of this type of misinterpretation that it is essential to move beyond in situ to operando. In catalysis, the term operando ideally means that there is some effort to simultaneously quantify and correlate reaction kinetics with catalyst structure. The approaches are not perfect and can be challenging, but they are better than simple in situ methods for the reasons given above. For example, it is important to remember that, in general, a simple measurement of conversion does not allow chemical kinetics to be determined. One must consider the chemical reaction engineering aspects of the reactor systems employed (e.g. heat and mass transport). As a pre-requisite to this work (and the ACS Catalysis paper), we have conducted a detailed finite element analysis of the microscope reaction cell used in our work. The model describes how to determine the reaction rate from a measurement of chemical conversion. This is employed in the current manuscript and enables us to estimate the turnover frequency (TOF) while observing the atomic structure of the catalyst. This makes our experiment truly

operando because we are able to link structure and structural dynamics with chemical kinetics over a wide range of different activity. This is not a trivial undertaking and is essential to correctly interpret the observation in terms of catalytic functionality. (The same complication of linking kinetics to conversion also applies to the TEM windowed cell reactors).

Regarding the specific citations mentioned by the reviewer. They are both important and interesting, but they are in situ TEM **not** operando TEM observations. The fluxional behavior reported by us in the paper by Li et al were performed in situ under reaction conditions, but there was no measurement of reaction kinetics during the TEM observations. In addition, the observations, though interesting, were not quantified (and were part of a large study employing a wide range of characterization tools to gain insight on structural changes that take in the more active in the water gas shift catalyst). The results reported on Au restructuring in a variety of gases are also interesting in situ observations – but there is no direct link established with catalysis. Because no operando measurements were performed (i.e. there were no measurements of chemical kinetics), it was not possible to establish a direct link between these behaviors and catalysis.

In the current work, we have explicitly and quantitatively correlated the reaction rate with the evolution of surface structure and several forms of dynamic structural processes. The Mars-van Krevelen mechanism is intimately associated with metal-support interfacial interactions, but in the current literature there are no direct observations of the interfacial behavior taking place during catalysis. The novelty of this work is that through operando TEM, we are now able to directly link the fluxional behavior associated with metal-support interaction with catalysis for the first time. This alone goes significantly beyond the two other works cited by the reviewer, and we think it qualifies the manuscript for publication in Nature Communications. In addition, we identify and characterize distinct types of fluxional dynamics and quantify each of these processes. We show that the degree of fluxionality is linearly related with turnover frequency (not temperature). These structural dynamics are integral to the Mars-van Krevelen mechanism, and to our knowledge, this has not been shown by any other group.

Changes to the text:

On page 5, we have added the following sentence to near the end of the introduction to emphasize the novelty of the contribution: “In the current work, we use *operando* TEM to explicitly and quantitatively correlate the catalytic reaction rate with two forms of structural dynamics processes. The Mars-van Krevelen mechanism is intimately associated with metal-support interactions, and we provide direct observations and characterization of the interfacial behavior over a range of different activities.”

Deleted the next two sentences in the same paragraph.

Deleted the next-to-last sentence in the same paragraph.

On page 33, we have re-written the first paragraph of the conclusion to more directly communicate the new catalysis and interface science revealed by our work: " In summary, we have employed aberration-corrected *operando* TEM to visualize the atomic-scale dynamic structural (i.e., fluxional) behavior occurring at and near Pt/CeO₂ metal-support interfaces caused by oxygen transfer during CO oxidation. Finite element modeling is performed to develop a reaction rate analysis that allows the catalyst's turnover frequency to be quantitatively correlated with different forms of structural dynamics processes. This analysis reveals a direct connection between activity and certain forms of fluxional behavior that are uniquely relevant to the catalytic reaction (not temperature). In this case, we show that the catalytically-relevant fluxional dynamics are associated with an enhanced rate of oxygen vacancy creation and annihilation on the surface at and close to the Pt perimeter sites, which leads to increased fluxional strain in the cation sublattice of the CeO₂ support and an enhanced overall reduction in the support surface. These results unambiguously demonstrate that there are dynamic and ongoing structural reconfigurations taking place around the metal-support interface which are correlated directly with catalysis.”

Deleted the next sentence in the same paragraph.

2. **How can one explain the particle fluxionality in the low activity state?** The particles should be covered by CO, but how does this lead to changes in coverage/surface energy that could induce fluxionality? As most of the Pt are inactive under such conditions (low CO conversion), **can beam effects really be excluded?**

Response: This is a very good question which touches on a topic of considerable interest. The catalyst may not be producing CO₂, but our observations show that significant surface chemistry is still taking place. Fluxionality

is caused by making and breaking chemical bonds on the surface of the catalyst - i.e. it is associated with surface chemistry in general and not just the rate limiting step for complete oxidation of CO. The detection of CO₂ in the gas phase implies that the rate limiting step has become activated, but there are other intermediate steps in the reaction pathway which are not rate limiting and will, by definition, have activation energies which are less than that for the rate limiting step. For this reason, these steps will be active at lower temperatures. In the case of CO oxidation via a Mars-van Krevelen mechanism, there are many steps in the reaction pathway leading to the eventual formation of the CO₂ product molecule. Important mechanistic steps include:

- Chemisorption of CO onto Pt nanoparticle
- Abstraction of interfacial O from the Pt/CeO₂ perimeter sites by chemisorbed CO species
- Reduction of molecular O₂ to back-fill oxygen vacancies

The first two steps take place at room temperature. We have several publications which provide evidence for this statement. Our recent ACS Nano publication (Atomic Scale Characterization of Fluxional Cation Behavior on Nanoparticles Surfaces: Probing Oxygen Vacancy Creation/Annihilation at Surface Sites, E. L. Lawrence, B. Levin, T. Boland, S. Chiang and P. A. Crozier, ACS Nano 2021 Vol. 15 Issue 2 Pages 2624-2634) shows that, even on pure CeO₂ at room temperature, oxygen vacancies are created and annihilated at special surface sites such as step edges, strained terraces, nano-sized facets, etc. These steps do not involve reduction of molecular oxygen (which is not activated on the bare CeO₂ surface at room temperature) but rather the abstraction of oxygen from surface lattice sites. (We do not comment on the form of the oxygen after abstraction from the surface lattice site). In the presence of Pt nanoparticles and CO gas, this process becomes considerably more active. This was demonstrated in our recent conference publication (Atomic-resolution Operando and Time-resolved In Situ TEM Imaging of Oxygen Transfer Reactions Catalyzed by CeO₂-supported Pt Nanoparticles, J. L. Vincent and P. A. Crozier, Microscopy and Microanalysis 26 (S2), 1694-1695). Here we showed that even at room temperature, Pt nanoparticles can show intense fluxional behavior in CO but not in N₂ gas. The destabilization of the Pt nanoparticle is a consequence of abstraction of oxygen from the metal-support interface sites. The abstracted oxygen is the same oxygen that binds the Pt particle to the CeO₂ support (via bridging Pt-O-Ce bonds). Thus, by removing this oxygen, the bond between the Pt particle and support is weakened and eventually fluxional behavior takes place. The effect is only observed in smaller particles (1 – 5 nm) because in larger particles the center of the interface between Pt and CeO₂ cannot be accessed by the CO. From this we conclude that the fluxional behavior that is observed before light-off is associated with more facile intermediate steps in the reaction. The fluxionality demonstrates that these intermediate reactions are going forward and backwards presumably due to the low activation energy. Since no significant amount of CO₂ is detected, we assume (as stated in the manuscript) that the CeO₂ fluxionality is associated with constant redistribution of oxygen vacancies at surface. Also, this dynamic bonding and debonding of adsorbates on the metal surface leads to distortions and strains, driving surface diffusion contributing to fluxionality in the Pt nanoparticles.

Significant quantities of CO₂ are only produced when the oxygen vacancies can be back-filled through the reduction of molecular oxygen and the Mars-van Krevelen catalytic cycle can be completed. When this happens, we see increased fluxional behavior indicating a greater rate of oxygen cycling at the surface. Interestingly, the outward relaxation also indicates that there is a higher concentration of oxygen vacancies when the catalyst starts to make CO₂. This implies that the most active form of the catalyst is associated with a higher average concentration of oxygen vacancies, while the high fluxionality indicates an enhanced rate of oxygen exchange and CO₂ production. All processes are present in the most active catalysts.

Finally, on electron beam effects, we have imaged this catalyst over several orders of electron beam dose rate (and magnification) under a variety of gas conditions. The fluxional behavior observed in the Pt nanoparticles is present even with electron dose rates as low as 60 e⁻/Å²/s, so we do not think the electron beam is significantly affecting the observations.

Changes to the text:

On page 20, we have added the following sentence to address the issue of electron beam effects: “We have imaged this catalyst over several orders of electron beam dose rate (and magnification) under a variety of gas conditions. The fluxional behavior observed in the Pt nanoparticles is present even at electron dose rates as low as 60 e⁻/Å²/s, so we do not think the electron beam is significantly affecting the observations.”

Starting on page 26, in the discussion section we have added the following text to deepen the discussion of fluxionality when the TOF is zero: “The fluxionality is caused by making and breaking chemical bonds on the surface of the catalyst, i.e. it is associated with surface chemistry and not just the rate limiting step for complete oxidation of CO. The detection of CO₂ in the gas phase implies that the rate limiting step has completed, but there are other intermediate steps in the reaction pathway which will have lower activation energies and will proceed at lower temperature. In the case of CO oxidation via a Mars-van Krevelen mechanism, there are many steps in the reaction pathway including:

- Chemisorption of CO onto Pt nanoparticle
- Abstraction of interfacial O from the Pt/CeO₂ perimeter sites by chemisorbed CO species
- Reduction of molecular O₂ to back fill oxygen vacancies

The first two steps can take place even at room temperatures. For example, we recently showed that, even on pure CeO₂ at room temperature, oxygen vacancies are created and annihilated at special surface sites such as step edges, strained terraces, nano-facets, etc. This process does not involve the reduction of molecular oxygen (which is not activated on the bare CeO₂ surface at room temperature), but rather the abstraction of oxygen from surface lattice sites. In the presence of Pt nanoparticles and in CO, this process becomes considerably more active. This was demonstrated in our recent conference publication (Vincent & Crozier, 2020), where we showed that, at room temperature, Pt nanoparticles can show intense fluxional behavior when CO is introduced but significantly less behavior in N₂ gas. The destabilization of the Pt nanoparticle is a consequence of abstraction of oxygen from the metal-support interface sites as discussed below.”

3. If yes, **what is the effect of reconstructions as those described by G. Ertl?**

Response: The work by G. Ertl delivered spectacular insight into the metastable structure-activity relationships for extended Pt (110) and Pt (111) single crystal surfaces performing CO oxidation. On an extended Pt single crystal surface, the oxidation of CO occurs between dissociatively chemisorbed O_{ad} and chemisorbed CO on the Pt surface. The reconstructions described by Ertl explain the chemical dynamics occurring during this process, which under most experimental conditions gave rise to oscillatory patterns that propagated with front speeds of ~μm/s. The relationship between such behavior of Pt single crystals and that of CeO₂-supported 1 – 2 nm Pt nanoparticles is not immediately clear. In the case of Pt/CeO₂, the CO oxidation mechanism involves metal-support interactions as well as a Mars-van Krevelen oxidation step at the perimeter of the Pt/CeO₂ metal/reducible-oxide interface. Additionally, the structural behavior of nanoparticles can differ significantly in comparison to extended surfaces due to the larger population of coordinatively unsaturated edge and corner sites. Energetics calculations of small (i.e., < 50 Å) metallic nanoparticles demonstrate that many structural isomers may exist with shallow kinetic barriers separating distinct configurations that show similar thermodynamic stability (Ajayan & Marks, 1988; Marks & Peng, 2016). At present, it is not readily apparent how such fluxionality should be understood within the chemical dynamics framework developed by Ertl, whose work involved oscillatory structural changes occurring over much larger length and time scales (and on non-supported metal surfaces), but it is an interesting matter of ongoing research.

Changes to the text:

On page 4 of the introduction, in the section on structural dynamics, added a citation to Ertl’s 2007 Nobel Prize lecture and to 1995 work on oscillatory kinetics on surfaces: (Ertl, G. (2007). Nobel Prize Lecture - Reactions at Surfaces: From Atoms to Complexity. <https://www.nobelprize.org/prizes/chemistry/2007/ertl/lecture/>) and (Imbihl, R. & Ertl, G. (1995). Oscillatory Kinetics in Heterogeneous Catalysis. *Chemical Reviews* 95, 697–733).

4. In addition to the experiments under constant CO/O₂ feed at increasing temperature, one could also change from inactive to active state by changing the CO:O₂ ratio. **Were such isothermal experiments (ruling out any temperature-induced fluxionality) also carried out?**

Response: We thank the reviewer for their comment. Such experiments were not carried out because the reaction rate is not strongly dependent on the partial pressures of the reactants. Additionally, changing the gas atmosphere may also affect the fluxionality of the catalyst, so for experiments performed under varied CO:O₂ ratio, there may still be some ambiguity in the interpretation (i.e., *is the fluxional behavior due to catalysis or due to the changing gas composition?*). However, the reviewer is correct to be concerned about decoupling thermal effects

from chemical ones, and for this reason we did conduct experiments to rule out any temperature-induced fluxionality. The results of these experiments are described in detail in Section 7 of the Supplemental Information (Figures S9 – S11). In short, the catalyst was imaged at 20 °C and at 300 °C in a non-reactive gas atmosphere of N₂. The observations reveal a notable lack of motion artifacts associated with fluxional behavior, as was observed under CO oxidation reaction conditions and during catalysis. Overall, the results of the experiments reported in the SI provide compelling evidence to support the assertion that the fluxional is not due to thermal affects but rather to catalytic surface chemistry.

Changes to the text:

On page 6 of the introduction, revised the sentence describing the control experiment performed to rule out temperature-induced fluxionality to make it clear that such experiments were carried out: “Additionally, the catalyst is observed at 300 °C in a non-reactive atmosphere of N₂ in order to rule out temperature-induced fluxionality from that associated with catalysis.”

5. With TOFs on the order of 1 per site-1 s-1 it seems that the 12.5 s time averages of Fig 4 rather capture the long-term fate of the catalyst, than the fluxionality connected to individual turnovers. **What aspect of the fluxionality is truly periodic in the sense of turnovers?**

Response: This is another excellent point from the reviewer. To minimize beam damage, we imaged the catalyst with a relatively weak electron beam intensity, which results in significant noise in the individual frames. The frame averaging approach that we used dramatically improves the signal to noise but also shows the presence of fluxional behavior due to blurring. This can be thought of as a motion artifact that one may see when taking a picture of something that is moving faster than the exposure time. In the present case, if the structure was stationary over 12.5 sec, then the frame-averaged images would appear sharp. The fact that the Pt nanoparticle and CeO₂ surface are blurred shows that the structure is dynamic within the 12.5 sec time resolution. For the analysis approach we employed here, the higher signal to noise present in the averaged image enables a more accurate quantification description of the fluxionality to be performed. However, the fluxionality is happening on a timescale that is comparable with the turnover frequencies, as shown Figure 2 and Supplementary Figure 12. These figures show changes in the Pt and CeO₂ surface structure with 0.5 s time resolution. The individual frames are noisy, but it is still clear that the fluxionality is cyclic on time frames that are comparable with the TOF.

Changes to the text:

On page 20, we have added the following sentences to address the issue of frame averaging: “We note that the signal-to-noise on individual 0.5 s frames is rather poor. The frame averaging approach improves the signal to noise but also shows the presence of fluxional behavior due to blurring and loss of contrast. For the analysis approach we employed here, the higher signal to noise present in the averaged image enables a more accurate quantification of the fluxionality. (However, the fluxionality is happening on a timescale that is comparable with the turnover frequencies, as shown Figure 2 and Supplementary Fig. 12 with 0.5 s time resolution).”

6. **How reliable is the finite elements prediction of turnovers** given the fluxionality of the particles, that would heavily influence the local reaction rates?

Response: We thank the reviewer for critically evaluating the manuscript and for asking another excellent question. The reaction rate that we measure under operando conditions is the steady-state average of the rate over the ensemble of all the particles in the ETEM. The finite element model provides a correction factor for this experimentally measured rate, which allows the reaction rate of particles imaged on the TEM grid to be determined quantitatively. The turnover frequency (TOF) that we calculate is determined by assuming that the activity is associated with the interfacial perimeter sites, the number of which are estimated as described in Section 2 of the Supplemental Information. It should be noted, however, that this is still an ensemble quantity that describes the average turnover rate for particles on the TEM grid. The precision of this TOF quantification is limited by the precision of the method used to determine the catalytic conversion, which is relatively small at the conditions where the catalyst behavior was quantified as shown by the error bars in Figure 3b.

We are fully aware of the limitations associated with this TOF description of activity, especially considering the fluxional behavior of the particles during the reaction, which may change the distribution and type of sites available on the catalyst surface and hence significantly influence the catalytic activity at any given instant in time. Overcoming this limitation would require one to determine the local reaction rate of individual nanoparticles (and ideally individual atomic surface sites) as they dynamically evolve in real time. To our knowledge this is not currently possible. An intermediate (yet still very insightful) level of detail may nonetheless be obtained by correlating the TOF with the local dynamic behavior of the particles and the surrounding support/interfacial area, which is demonstrated for the first time in the present work. This approach circumvents the difficulty associated with determining the local reaction rate of specific nanoparticles yet still allows one to clearly identify dynamic structural behavior that is intimately connected with the functioning of the catalytic mechanism.

Changes to the text:

On page 19, in the paragraph describing the derivation of the TOF values, we have added the phrases “ensemble” and “average” before *in situ* TOF in order to emphasize the fact that the turnover values are describing the average turnover rate of the ensemble of particles supported on the TEM grid.

Additional Formatting Revisions Implemented in the Revised Manuscript:

Throughout the manuscript, a number of changes were made to comply with the re-submission formatting instructions as outlined on the Nature Communications Guide to Formatting Articles.

Changes to the text:

On every page, added page numbers to each page.

On page 1, revised the title to comply with the Nature Communications title formatting requirements: “Atomic Level Fluxional Behavior and its Impact on Activity: CO Oxidation over CeO₂-supported Pt Catalysts.” Made the same change to the title page of the SI.

On page 2, under guide of formatting requirements, the length of abstract was reduced to fewer than 150 words.

Shortened multiple Methods sub-section headings to fewer than 60 words:

- Page 6, shortened subheading to: “Catalyst Preparation and Structural/Activity Characterization.”
- Page 10, shortened subheading to: “Finite Element Simulation of Operando ETEM Reactor.”
- Page 10, shortened subheading to: “In Situ ETEM in Non-reactive Gases at Elevated Temperature.”

Removed numbering for Methods subheadings, as well as all references to them in the text.

Shortened multiple Results sub-section headings to fewer than 60 words:

- Page 16, shortened subheading to: “Quantifying the Catalytic Reaction Rate in the ETEM Reactor.”
- Page 19, shortened subheading to: “Fluxional Dynamics Occurring During Catalysis.”

Removed numbering for Results subheadings, as well as all references to them in the text.

Changed the reference formatting to styles described in the formatting guide. Throughout the text, removed spaces separating text and in-text numbered citations.

On pages 41 – 46, the figure captions were revised in order to follow the title and legend formatting requirements outlined in style guide.

Changed all in-text references to Supplementary information, figures, tables, and so on, to match that outlined in the style guide. Formatting the SI to follow the requirements described in the style guide.

Expressed all inverse unit dimensions with negative integers.

REVIEWERS' COMMENTS

Reviewer #1 (Remarks to the Author):

The authors have taken into account the comments and criticisms raised by the referees in their reports. The paper can now be accepted for publication.

Reviewer #2 (Remarks to the Author):

The authors have answered all questions/comments very well and several aspects should now be even clearer to the readers. I would have many more questions though ;-)